# *N*-(2-Arylmethylthio-4-Chloro-5-Methylbenzenesulfonyl)amide Derivatives as Potential Antimicrobial Agents—Synthesis and Biological Studies

**DOI:** 10.3390/ijms21010210

**Published:** 2019-12-27

**Authors:** Beata Żołnowska, Jarosław Sławiński, Katarzyna Garbacz, Małgorzata Jarosiewicz, Anna Kawiak

**Affiliations:** 1Department of Organic Chemistry, Faculty of Pharmacy, Medical University of Gdańsk, Al. Gen. J. Hallera 107, 80-416 Gdańsk, Poland; 2Department of Oral Microbiology, Faculty of Medicine, Medical University of Gdańsk, Dębowa 25, 80-204 Gdańsk, Poland; katarzyna.garbacz@gumed.edu.pl (K.G.); malgorzata.jarosiewicz@gumed.edu.pl (M.J.); 3Department of Biotechnology, Intercollegiate Faculty of Biotechnology, University of Gdańsk and Medical University of Gdańsk, Ul. Abrahama 58, 80-307 Gdańsk, Poland; anna.kawiak@biotech.ug.edu.pl

**Keywords:** benzenesulfonamide, antibacterial, biofilm, synthesis, cytotoxicity, MRSA

## Abstract

Rising resistance of pathogenic bacteria reduces the options of treating hospital and non-hospital bacterial infections. There is a need to search for newer chemotherapies that will show antimicrobial ability against planktonic cells as well as bacterial biofilms. We have synthesized a series of *N*-(2-arylmethylthio-4-chloro-5-methylbenzenesulfonyl)amides, namely, molecular hybrids, which include a 2-mercaptobenzenosulfonamide fragment and either cinnamic or cyclohexylpropionic acid residues. The antimicrobial activity of compounds **8**–**17** was evaluated on Gram-positive, Gram-negative bacteria and fungal species. Experiments took into account investigation of antibacterial activity against planktonic cells as well as biofilms. Compounds **8**–**17** showed high bacteriostatic activity against staphylococci, with the most active molecules **10** and **16** presenting low MIC values of 4–8 μg/mL against reference methicillin-resistant *Staphylococcus aureus* (MRSA) and methicillin-sensitive *S. aureus* (MSSA) strains as well as clinical isolates. Compounds **10** and **16** also showed an ability to inhibit biofilms formed by MRSA and MSSA. The potential of **10** and **16** as antibiofilm agents was supported by cytotoxicity assays that indicated no cytotoxic effect either on normal cells of human keratinocytes or on human cancer cells, including cervical, colon, and breast cancer lines.

## 1. Introduction

An increasing resistance of pathogenic bacteria towards existing antibiotics is a global therapeutic problem, affecting both hospital and non-hospital infections. In 2018, the rise of resistant strains was considered by the World Health Organization (WHO) to be a global concern [1]. One of the main causes of antibiotic resistance is abuse and misuse of drugs by patients. Use of huge amounts of antibiotics over the past 75 years has made almost all pathogenic bacteria resistant to common antibiotics [2]. As a result of this process, the number of multidrug resistant strains, insensitive to commercial treatments, is still growing [3]. Therefore, rising resistance reduces the options of treating bacterial infections in hospital interventions, including surgery, transplantation, and chemotherapy. However, the resistance mentioned above is not completely explicated by the rise of resistant strains but also by the formation of biofilms by bacteria.

Bacterial biofilm is an organized multicellular structure adjacent to solid surfaces, formed by cells of microbes of one or more species immersed in a biofilm matrix—An exopolymer matrix consisting of polysaccharides, proteins, lipids, and DNA [4]. The matrix function is to serve as a scaffold for the three-dimensional construction of the biofilm as well as to provide protection to the cells in this structure. Compared to planktonic cells, microorganisms growing in the biofilm population are less sensitive to antibiotics and the defense mechanisms of the human body. It has been reported that biofilms exhibit a thousand times more resistance than planktonic bacteria to antibacterial agents [5]. High antimicrobial resistance and tolerance has been associated with the protective role of the biofilm matrix acting as a barrier against antibiotic penetration. Moreover, slow growth rate, altered metabolism, persister cells and oxygen gradients are thought to be engaged in biofilm tolerance and resistance [6]. Since biofilm formation is considered to be responsible for failure of treatment with antibiotics, and 65–80% of all infections are thought to be related to biofilm [7], scientists have searched for newer and newer chemotherapies that will show the ability to fight these biofilms.

*Staphyloccocus aureus* is a major human pathogen which causes skin, soft tissue, respiratory, and blood stream infections. It is responsible for a variety of diseases including mild infections or food poisoning as well as severe infections such as osteomyelitis, endocarditis, and toxic shock syndrome. Mortality due to methicillin-resistant *Staphylococcus aureus* (MRSA) is the leading cause of death due to bacterial infections [8]. Furthermore, *Staphyloccocus aureus* is the leading clinical problem in biofilm-related illnesses such as periodontitis, ocular infections, and chronic wound infection [9].

In our previous reports, we proved that 4-chloro-2-mercapto-5-methylbenzenesulfonamides show a wide range of antibacterial activity against numerous bacterial strains obtained from patients with infections of the oral cavity: gingivitis, periodontal diseases, corrosive ulcers, stomatitis (swabs, paper point), respiratory tract (sputum, swabs), or intestinal tract (stool) [10,11]. Our promising results prompted us to search for new low-molecular antibacterial agents among the 4-chloro-2-mercapto-5-methylbenzenesulfonamides. In this work, we developed a series of *N*-(2-arylmethylthio-4-chloro-5-methylbenzenesulfonyl)amides, applying the hybridization strategy, which is a well-known approach for designing new drugs with novel pharmacological properties and potential to prevent resistance. Molecular hybrids were obtained by combining the 2-mercaptobenzenosulfonamide fragment with a cinnamic acid residue. The antimicrobial activity of cinnamic acid has been well documented in several reviews [12,13]. Cinnamic acids have also been used by scientists to modify parameters such as the potency, permeability and solubility of a selected drug or pharmacophore. Within the structure of our compounds, we also incorporated a cyclohexylpropionic acid residue as an aliphatic analogue of cinnamic acid to investigate the impact of a saturated versus an aromatic system on antibacterial activity.

The aim of the present study was to evaluate a group of *N*-(2-arylmethylthio-4-chloro-5-methylbenzenesulfonyl)amides as potential antimicrobial agents exhibiting bacteriostatic/bactericidal activity against planktonic bacterial cells as well as antibiofilm activity against structures formed by some strains associated with biofilm-related diseases.

## 2. Results and Discussion

### 2.1. Chemistry

The synthetic route adopted for the preparation of the target *N*-(2-arylmethylthio-4-chloro-5-methylbenzenesulfonyl)amides is shown in Scheme 1. The starting *N*-(benzenesulfonyl)cyanamide potassium salts **3**–**7** were prepared by reaction of 3-amino-1,1-dioxo-1,4,2-benzodithiazine (**1**) with anhydrous K_2_CO_3_ and alkylating agents according to reported procedures [14,15]. For the synthesis of novel *N*-(2-arylmethylthio-4-chloro-5-methylbenzenesulfonyl)amide derivatives **8**–**12**, the convenient acylation of sulfonamides was used [16]. The *N*-(2-arylmethylthio-4-chloro-5-methylbenzenesulfonyl)-3-cyclohexylpropanamides **8**–**12** were obtained by heating monopotassium salts **3**–**7** with cyclohexylpropanoic acid at 115 °C for 1–2 h, with 70–85% yield (Scheme 1). Heating of a mixture of monopotassium salts **3**–**7** with cinnamic acid in water at 120 °C for 96 h led to the N-(2-arylmethylthio-4-chloro-5-methylbenzenesulfonyl)cinnamamides **13**–**17** with a good yield 55–87% (Scheme 1). All reactions were monitored by thin layer chromatography (TLC).

New structures were confirmed by mass spectrometry and spectroscopic data (IR, ^1^H NMR, ^13^C NMR). The carbonyl group was identified in the IR spectra as a characteristic C=O absorption band at 1638–1734 cm^−1^ and in the ^13^C NMR spectra as carbon signal at 163.4–180.9 ppm. In addition, the appearance of a signal at 12.20–12.82 ppm in the ^1^H NMR spectra indicated the presence of the SO_2_NHCO moiety. All compounds were also characterized by elemental analyses (C, H, N).

### 2.2. Antimicrobial Activity Agianst Reference Microbial Strains

The antimicrobial activity of compounds **8**–**17** was evaluated on Gram-positive bacteria: *Staphylococcus aureus* ATCC 6538 (MSSA), *Staphylococcus aureus* ATCC 43300 (MRSA), *Staphylococcus epidermidis PCM* 2118, *Enterococcus faecalis* ATCC 11420, Gram-negative bacterial strains: *Escherichia coli ATCC* 11229, *Pseudomonas aeruginosa* ATCC 15442, and *Proteus vulgaris* NCTC 4635, as well as on one fungal species: *Candida albicans* ATCC 10231. The tests were carried out using a serial dilution method, allowing the determination of MIC (minimal concentration inhibiting bacterial growth), MBC (minimal bactericidal concentration), and MFC (minimal fungicidal concentration) values. Results of the experiments are presented in Table 1, Table 2 and Table 3.

The MIC value describes an effect of a chemotherapeutic agent on a microorganism and quantitatively represents a potency of a given substance. This parameter expresses the bacteriostatic effect of the drug, which involves inhibiting a growth of bacteria; however, this action does not lead to the death of existing microorganisms.

Another parameter describing quantitatively the potency of an active substance is the MBC value, referring to the minimal concentration of antibiotic at which the number of bacteria capable of producing colonies decreases to zero. This parameter defines the bactericidal effect of the chemotherapeutic, causing not only inhibition of bacterial colony growth but also leading to the death of already existing bacterial cells. Usually, the MBC value is higher than the MIC. A high value of the MBC/MIC ratio indicates microorganism tolerance of a given substance. It means that the microorganism is resistant to the bactericidal effect of the compound; however, its growth is inhibited.

The results indicated that compounds **8**–**17** showed high bacteriostatic activity against Gram-positive reference strains, especially staphylococci. It is worth pointing out that the majority of compounds had an ability to inhibit Gram-positive growth at low concentrations from 4 μg/mL to 8 μg/mL. *E. faecalis* also exhibited susceptibility towards some of the tested compounds (MIC: 8–32 μg/mL), except for **8**, **13** and **17** (Table 1). The activity of compounds **8**–**17** against Gram-negative bacteria turned out to be low, with MIC values ranging from 64 μg/mL to >128 μg/mL (Table 2). Low activity can be explained by different structure of the cell wall of Gram-negative and Gram-positive bacteria. The cellular wall of Gram-negative species is composed of a thick lipoprotein-lipopolysaccharide layer which shields a thin layer of peptidoglycan. On the other hand, Gram-positive species have a cell wall composed of dozens of peptidoglycan layers that may be susceptible to the tested *N*-benzenesulfonylamides.

Among the tested compounds, one, *N*-(benzenesulfonyl)-3-cyclohexylpropanamide **10** (R = 3-CF_3_Ph), was identified as the most active agent against all Gram-positive species. It presented a low MIC value of 8 μg/mL as well as low or medium MBC values (8–64 μg/mL). In addition, compound **10** exhibited a low MBC/MIC ratio against MRSA and *S. epidermidis*, which indicates a negligible tolerance of microorganisms to this substance (Table 3). A satisfactory MBC/MIC ratio equals 1 against *S. epidermidis* was also noticed for *N*-(benzenesulfonyl)cinnamamide **15** (R = 3-CF_3_Ph), whereas the MIC and MBC values obtained for this compound against S. epidermidis were 8 μg/mL. It should be noted that *N*-benzenesulfonylamide derivatives **9**–**10** and **14**–**15**, with 3- and 4-trifluoromethylphenyl substituents, were also characterized by having strong bactericidal activity against *S. epidermidis*, MBC values for these compounds ranged between 8 and 16 μg/mL.

Considering bactericidal aspects, it is evident that *N*-(benzenesulfonyl)cinnamamide **16** containing the 1-naphthyl substituent is the most potent agent killing *S. epidermidis* (MBC = 4 μg/mL). Changing the 1-naphthyl substituent in compound **16** into the isomeric 2-naphthyl (**17**) dramatically reduced its bactericidal activity, whereas bacteriostatic activity was preserved. Interestingly, compound **16** showed outstanding bacteriostatic activity against all tested Gram-positive strains with MIC values ranging from 4 to 8 μg/mL.

Compounds **8**–**17** exhibited weak antifungal activity; their MIC and MFC values were 64 μg/mL and 128 μg/mL, respectively.

Results of the tests did not indicate significant impact of either aromatic or saturated systems attached to the sulfonamide moiety on antibacterial activity.

### 2.3. Antibacterial Activity Against Clinical Methicillin-Resistant and Methicillin-Sensitive S. aureus Strains

The most promising compounds **10** and **16** were evaluated for bacteriostatic and bactericidal activity against a panel of clinical isolates of methicillin-resistant *S. aureus* (50–54) and methicillin-sensitive *S. aureus* (55–59). Both MRSA and MSSA strains were derived from various human infections. MIC and MBC values are presented in Table 4.

Test results showed high bacteriostatic activity of both compounds **10** and **16**, characterized by MIC = 4 µg/mL for **10** and MIC: 4–8 µg/mL for compound **16**. It is worth noting that strains isolated from clinical specimens were 2-fold more sensitive (MIC = 4 µg/mL, Table 4) to bacteriostatic activity of **10** than reference strains (MIC = 8 µg/mL, Table 1). Conducted experiments have indicated no bactericidal activity of compounds **10** and **16** against clinical isolates (MBC > 128 µg/mL).

Different susceptibility of strains within one species can be explained by genetic variances between laboratory reference strains and clinical strains [17]. In the context of searching for antibiotics, particularly effective in the treatment of nosocomial bacterial infections, the high potential of investigated molecules **10** and **16** is noteworthy.

These studies showed that both MSSA and MRSA strains exhibited similar susceptibility to the analyzed compounds **10** and **16**. It should be stressed that antibiotic therapy of MRSA infections is usually more difficult and demanding than those caused by MSSA. Methicillin-resistant *S. aureus* (MRSA) are insusceptible to all beta-lactams, including antibiotics with beta-lactamase inhibitors and carbapenems. Moreover, resistance to methicillin accompanies the resistance to other groups of antibiotics (macrolides, lincosamides, aminoglycosides, fluoroquinolones, and sulfonamides), and determines multiple drug resistance [18].

### 2.4. Inhibition of Biofilm Formation

The influence of compounds **10** and **16** on biofilm formation (Figure 1) was studied using four *S. aureus* strains being strong biofilm producers including two MRSA (51, 52) and two MSSA (58, 59) strains. For many decades, *S. aureus* has been identified as the most common etiologic agent of device related infections, associated with biofilm formation. These infections are hard to cure because the specific structure of biofilm confers resistance to the host immune response and to the action of antibiotics [19].

The highest sensitivity to compound **10** was demonstrated by MSSA 58 biofilm. In this case, biofilm formation was evidently inhibited at a concentration of 2 μg/mL, which corresponds to ½ MIC (67% inhibition at ½ MIC). Application of a concentration equal to the MIC resulted in inhibition of biofilm growth in about 91%. However, strong inhibition of MSSA 59 biofilm production was observed at 4× MIC (91% inhibition). MRSA 51 and 52 biofilms were also susceptible to compound **10**, especially at concentration of 4 × MIC (88–89% inhibition).

Results obtained for compound **16** showed that inhibition of biofilm MRSA 51 was observed at a concentration equal to the MIC (35% inhibition). On the other hand, a strong effect on MRSA 51 and 52 as well as MSSA 59 inhibition (87–90% inhibition) occurred at concentration of 16 µg/mL, which equals 4 × MIC. Compound **16** turned out to be good antibiofilm agent, inhibiting MSSA 58 biofilm formation at MIC concentration (44% inhibition at 8 µg/mL). Applying **16** at a concentration of 2 × MIC led to an inhibition of 86%.

An ability to inhibit biofilm by **10** and **16** may find application in the development of antibiofilm agents, as well as helping to overcome the problem of colonization by staphylococci on implantable medical devices or injured host tissue, especially in healthcare settings. Biofilm-associated infections have been associated with devices ranging from contact lenses to prosthetic heart valves, cardiac pacemakers, cerebrospinal fluid shunts, joint replacements, implanted catheters, and intravascular lines. They also have a substantial impact on patient morbidity and mortality [19].

### 2.5. Cytotoxic Activity

To assess if the influence displayed against bacteria could be associated with selective toxicity or a broad toxic influence, we performed an assay on the human keratinocyte (HaCaT cell line) as well as on three human cancer cell lines: HCT-116 (colon cancer), HeLa (cervical cancer), and MCF-7 (breast cancer). Analysis was performed using the MTT test and 72 h of incubation. The calculated IC_50_ values (the concentration required for 50% inhibition of cell viability) were shown in Table 5.

The results of the experiments showed that the tested compounds do not exhibit cytotoxic activity against cancer cells (IC_50_ HCT-116: 66–86 µM; IC_50_ HeLa: 42–165 µM; IC_50_ MCF-7: 65–96 µM); more importantly, they do not show cytotoxicity toward normal cells at concentrations corresponding to the MIC values. The IC_50_ value equals 95 µg/mL for the most active compound **10** against HaCaT cells, which was higher than the MBC values (MBC: 8–64 µg/mL) in which the compound shows a bacteriostatic effect against all Gram-positive bacteria (methicillin-sensitive *S. aureus*, MRSA, *S. epidermidis* and *E. faecalis*). Similar observations were noticed for **9**, **10**, **14**–**16**, the most effective bactericidal agents against S. epidermidis (MBC: 4–16 µg/mL, HaCaT IC_50_: 19–147 µg/mL).

## 3. Materials and Methods

### 3.1. Synthesis

Melting points were measured using Stuart SMP30 (Bibby Scientific Limited, Stone Staffordshire UK) apparatus and were uncorrected. IR spectra were recorded on Nicolet iS5 FTIR spectrometer (Thermo Fisher Scientific, Waltham, MA, USA) in KBr pellets; the absorption range was 400–4000 cm^−1^. ^1^H NMR and ^13^C NMR spectra were obtained on Varian Unity Plus 500 apparatus (Varian, Palo Alto, CA, USA). Chemical shifts are reported in parts per million (ppm). Moreover, resonance multiplicity is presented as: s (singlet), d (doublet), t (triplet), q (quartet), and m (multiplet). Elemental analyses were obtained on PerkinElmer 2400 Series II CHN Elemental Analyzer apparatus (PerkinElmer, Shelton, CT, USA) and the results indicated by the symbols of the elements were within ±0.4% of the theoretical values. Thin-layer chromatography (TLC) was conducted on Merck Kieselgel 60 F254 plates (Merck, Darmstadt, Germany) and visualized with UV. Column chromatography was performed using silica gel with pore size 60 Å, 220–440 mesh particle size, 35–75 μm particle size and a mixture of ethyl acetate/*n*-hexane (Sigma-Aldrich Co., St. Louis, MO, USA) as an eluent. Purity of compounds was analyzed by RP-HPLC (Model LC-10AD, Shimadzu) (Shimadzu USA Manufacturing INC. Canby, OR, USA), Column: Gemini 4.6 × 250 mm; C6-phenyl; 5 µm; 110 Å, Mobile Phase: A—grade water with 0.1% (*v*/*v*) trifluoroacetic acid (Sigma-Aldrich Co., St. Louis, MO, USA) B—80% acetonitrile-water containing 0.08% (*v*/*v*) trifluoroacetic acid (Avantor Performance Materials Poland S.A., Gliwice, Poland), Linear gradient 5–100% B in 45 min, Flow Rate: 1 mL/min. The purity of compounds, determined by RP-HPLC, was >95%. High resolution mass spectrometry (HRMS) were conducted on TripleTOF 5600+ mass spectrometer (AB SCIEX, Framingham, MA, USA) equipped with a DuoSpray^TM^ Ion Source and coupled with Micro HPLC system Ekspert™ microLC 200 (Eksigent Redwood City, CA, USA); Column: HALO Fused-Core C18 (50 × 0.5 mm, 2.7 μm) (Eksigent), thermostated at 50 °C; Flow: 30 µL/min; Mobile Phase: A: 0.1% formic acid in water, B: 0.1% formic acid in acetonitrile; Isocratic program 100% B, 4 min.

The following starting compounds were obtained according to the reported methods: 3-amino-6-chloro-7-methyl-1,1-dioxo-1,4,2-benzodithiazine (**1**) [14], *N*-(4-chloro-2-mercapto-5-methylbenzenesulfonyl)cyanamide dipotassium salt (**2**) [14], *N*-(2-benzylthio-4-chloro-5-methylbenzenesulfonyl)cyanamide potassium salt (**3**) [14], *N*-{4-chloro-5-methyl-2-[(4-trifluoromethylbenzyl)thio]benzenesulfonyl}cyanamide potassium salt (**4**) [15], *N*-{4-chloro-5-methyl-2-[(3-trifluoromethylbenzyl)thio]benzenesulfonyl}cyanamide potassium salt (**5**) [15], *N*-[4-chloro-5-methyl-2-(naphthalen-1-ylmethylthio)benzenesulfonyl]cyanamide potassium salt (**6**) [15], and *N*-[4-chloro-5-methyl-2-(naphthalen-2-ylmethylthio)benzenesulfonyl]cyanamide potassium salt (**7**) [16].

#### 3.1.1. General Procedure for the Synthesis of *N*-(2-arylmethylthio-4-chloro-5-methylbenzenesulfonyl)-3-cyclohexylpropanamides (**8**–**12**)

A solution of *N*-(benzenesulfonyl)cyanamide monopotassium salt **3**–**7** (0.7 mmol) in 3-cyclohexylpropanoic acid (4.1 mL) was stirred at 115 °C for 1–2 h. After the reaction, water (5 mL) was added to the reaction mixture. The resulting oil was dissolved in ethyl acetate (100 mL) and the organic layer was then washed with 10% aqueous NaHCO_3_ (3 × 30 mL), saturated aqueous NaCl (2 × 30 mL) and dried with MgSO_4_. The solvent was evaporated to dryness, giving an oily product that was purified as indicated below.

##### *N*-(2-Benzylthio-4-chloro-5-methylbenzenesulfonyl)-3-cyclohexylpropanamide (**8**)

Starting with **3** (0.274 g) and conducting reaction for 1.5 h, the title compound (**8**) was obtained (0.261 g, 80%). Melting point and spectroscopic analysis were in accordance with the previously described in [16]. HRMS (ESI-TOF) *m/z* calculated for [M + H]^+^ 466.1277, found [M + H]^+^ 466.1273.

##### *N*-[4-Chloro-5-methyl-2-(4-trifluoromethylbenzylthio)benzenesulfonyl]-3-cyclohexylpropanamide (**9**)

Starting with **4** (0.321 g) and conducting reaction for 2 h, the title compound (**9**) was obtained after previous crystallization from ethanol (0.318 g, 85%): m.p. 148.0–150.8 °C; t_R_ = 47.92 min; IR (KBr): 3108 (NH); 2925 (CH_3_); 2854, 2844 (CH_2_); 1682 (C=O); 141 (C=C); 1324, 1111 (SO_2_) cm^−1^; ^1^H NMR (500 MHz, DMSO-*d*_6_): δ 0.76 [q, 2H, aliphatic], 1.04–1.13 [m, 4H, aliphatic], 1.255 [q, 2H, aliphatic], 1.54–1.62 [m, 5H, aliphatic], 2.16 [t, 2H, aliphatic], 2.34 [s, 3H, CH_3_], 4.53 [s, 2H, CH_2_S], 7.63 [d, 2H, arom], 7.66 [s, 1H, H-3], 7.68 [d, 2H, arom], 7.90 [s, 1H, H-6], 12.28 [s, 1H, NH] ppm; ^13^C NMR (125 MHz, DMSO-*d*_6_): δ 19.4, 26.1, 26.5, 31.7, 32.8, 33.3, 35.9, 36.8, 125.8, 125.8, 128.6, 130.3, 133.2, 134.1; 135.2, 135.6, 139.4, 141.7, 172.1 ppm. Anal. C_24_H_27_ClF_3_NO_3_S_2_ (534.05); C%: 53.98; H%: 5.10; N%: 2.62, found: C%: 53.98; H%: 5.09; N%: 2.66. HRMS (ESI-TOF) *m/z* calculated for [M + H]^+^ 534.1151, found [M + H]^+^ 534.1153.

##### *N*-[4-Chloro-5-methyl-2-(3-trifluoromethylbenzylthio)benzenesulfonyl]-3-cyclohexylpropanamide (**10**)

Starting with **5** (0.321 g) and conducting reaction for 1 h, the title compound (**10**) was obtained after previous purification on silica gel using AcOEt/*n*-hexane (*v*/*v* = 1:4) as an eluent (0.262 g, 70%): m.p. 90.2–92.7 °C; t_R_ = 47.99 min; IR (KBr): 3216 (NH); 3065, 3081 (CH arom); 2986, 2924 (CH_3_); 2850 (CH_2_); 1720 (C=O); 1436 (C=C); 1330, 1128 (SO_2_) cm^−1^; ^1^H NMR (500 MHz, DMSO-*d*_6_): δ 0.78 [q, 2H, aliphatic], 1.00–1.20 [m, 4H, aliphatic], 1.27 [q, 2H, aliphatic], 1.52–1.68 [m, 5H, aliphatic], 2.18 [q, 2H, aliphatic], 2.33 [s, 3H, CH_3_], 4.53 [s, 2H, CH_2_S], 7.56 [t, 1H, arom], 7.63 [d, 1H, arom], 7.65 [s, 1H, H-3], 7.71 [d, 1H, arom], 7.77 [s, 1H, arom], 7.90 [s, 1H, H-6], 12,3 [s, 1H, NH] ppm; Anal. C_24_H_27_ClF_3_NO_3_S_2_ (534.05); C%: 53.98; H%: 5.10; N%: 2.62, found: C%: 53.58; H%: 5.06; N%: 2.56. HRMS (ESI-TOF) *m/z* calculated for [M + H]^+^ 534.1151, found [M + H]^+^ 534.1161.

##### *N*-[4-Chloro-5-methyl-2-(naphthalen-1-ylmethylthio)benzenesulfonyl]-3-cyclohexylpropanamide (**11**)

Starting with **6** (0.309 g) and conducting reaction for 2 h, the title compound (**11**) was obtained after previous crystallization from ethanol (0.300 g, 83%): m.p. 122.6–125.0 °C; t_R_ = 47.08 min; IR (KBr): 3203 (NH); 3066, 3086 (CH arom); 2919 (CH_3_); 2849 (CH_2_); 1693 (C=O); 1448 (C=C); 1358, 1172 (SO_2_) cm^−1^; ^1^H NMR (500 MHz, DMSO-*d*_6_): δ 0.75 [q, 2H, aliphatic], 0.98–1.08 [m, 4H, aliphatic]; 1.25 [q, 2H, aliphatic], 1.50–1.65 [m, 5H, aliphatic], 2.06 [t, 2H, aliphatic], 2.37 [s, 3H, CH_3_], 4.90 [s, 2H, CH_2_S], 7.46 [t, 1H, arom], 7.53–7.59 [m, 2H, arom], 7.60 [d, 1H, arom], 7.76 [s, 1H, H-3], 7.90 [d, 1H, arom], 7.92 [s, 1H, H-6], 7.96 [d, 1H, arom], 8.23 [d, 1H, arom], 12.14 [s, 1H, NH] ppm. Anal. C_27_H_30_ClNO_3_S_2_ (516.12); C%: 62.83; H%: 5.86; N%: 2.21, found: C%: 62.64; H%: 5.61; N%: 2.75. HRMS (ESI-TOF) *m/z* calculated for [M − H]^−^ 514.1283, found [M − H]^−^ 514.1244.

##### *N*-[4-Chloro-5-methyl-2-(naphthalen-2-ylmethylthio)benzenesulfonyl]-3-cyclohexylpropanamide (**12**)

Starting with **7** (0.309 g) and conducting reaction for 1.5 h, the title compound **12** was obtained after previous purification on silica gel using AcOEt/*n*-hexane (*v*/*v* = 1:4) as the eluent (0.260 g, 72%): *m.p* 157.5–158.8 °C; t_R_ = 48.56 min; IR (KBr): 3236 (NH); 3061 (CH arom); 2922 (CH_3_); 2850 (CH_2_); 1684 (C=O); 1449 (C=C); 1357, 1170 (SO_2_) cm^−1^; ^1^H NMR (500 MHz, DMSO-*d*_6_): δ 0.75 [q, 2H, aliphatic], 1.00–1.15 [m, 4H, aliphatic], 1.25 [q, 2H, aliphatic], 1.50–1.65 [m, 5H, aliphatic], 2.17 [t, 2H, aliphatic], 2.33 [s, 3H, CH_3_], 4.59 [s, 2H, CH_2_S], 7.48–7.56 [m, 2H, arom], 7.56 [s,1H, arom], 7.72 [s, 1H, H-3], 7.81–7.85 [m, 1H, arom], 7.86 [s, 1H, arom], 7.88 [d, 2H, arom], 7.93 [s, 1H, H-6], 12.27 [s, 1H, NH] ppm. Anal. C_27_H_30_ClNO_3_S_2_ (516.12); C%: 62.37; H%: 5.67; N%: 2.73, found: C%: 62.74; H%: 5.83; N%: 2.72. HRMS (ESI-TOF) *m/z* calculated for [M − H]^−^ 514.1283, found [M − H]^−^ 514.1272.

#### 3.1.2. General Procedure for the Synthesis of *N*-(2-arylmethylthio-4-chloro-5-methylbenzenesulfonyl)cinnamamides (**13**–**17**)

A mixture of *N*-(benzenesulfonyl)cyanamide monopotassium salt **3**–**7** (0.7 mmol) and cinnamic acid (1.4 mmol) in water (6 mL) was stirred in a pressure tube at 120 °C for 96 h. The product was filtered off from the hot reaction mixture and purified as indicated below.

##### *N*-{[2-(Benzylthio)-4-chloro-5-methylphenyl]sulfonyl}cinnamamide (**13**)

Starting with **3** (0.274 g), the title compound (**13**) was obtained after previous crystallization from ethanol (0.250 g, 78%). Melting point and spectroscopic analysis were in accordance with the previously described in [16]. HRMS (ESI-TOF) *m/z* calculated for [M + H]^+^ 458.0651, found [M + H]^+^ 458.0657.

##### *N*-[4-Chloro-5-methyl-2-(4-trifluoromethylbenzylthio)benzenesulfonyl]cinnamamide (**14**)

Starting with **4** (0.321 g), the title compound (**14**) was obtained after previous crystallization from ethanol (0.320 g (87%): m.p. 191.8–192.5 °C; t_R_ = 45.20 min; IR (KBr): 3210 (NH); 3064 (CH arom); 2922 (CH_3_); 2855 (CH_2_); 1678 (C=O); 1619, 1452 (C=C); 1326, 1111 (SO_2_) cm^−1^; ^1^H NMR (500 MHz, DMSO-*d*_6_): δ 2.35 [s, 3H, CH_3_], 4.54 [s, 2H, CH_2_S], 6.73 [d, 1H, CH=], 7.42–7.48 [m, 5H, arom], 7.52 [d, 1H, CH=], 7.68 [s, 1H, H-3], 7.85 [d, 4H, arom], 7.96 [s, 1H, H-6], 12.53 [s, 1H, NH] ppm. Anal. C_24_H_19_ClF_3_NO_3_S_2_ (525.99); C%: 54.56; H%: 3.56; N%: 2.72, found: C%: 54.62; H%: 3.54; N%: 2.72. HRMS (ESI-TOF) *m/z* calculated for [M + H]^+^ 526.0525, found [M + H]^+^ 526.0442.

##### *N*-[4-Chloro-5-methyl-2-(3-trifluoromethylbenzylthio)benzenesulfonyl]cinnamamide (**15**)

Starting with **5** (0.321 g), the title compound (**15**) was obtained after previous crystallization from ethanol (0.261 g, 71%): m.p. 188.3–189.5 °C; t_R_ = 45.47 min; IR (KBr): 3192 (NH); 3069 (CH arom); 2921 (CH_3_); 2864 (CH_2_); 1680 (C=O); 1618, 1451 (C=C); 1383, 1183 (SO_2_ sym) cm^−1^; ^1^H NMR (500 MHz, DMSO-*d*_6_): δ 2.35 [s, 3H, CH_3_], 4.52 [s, 2H, CH_2_S], 6.68 [d, 1H, CH=], 7.36 [t, 1H, arom], 7.42–7.48 [m, 3H, arom], 7.50 [d, 1H, CH=], 7.53–7.59 [m, 3H, arom], 7.66 [d, 2H, arom], 7.75 [s, 1H, H-3], 7.97 [s, 1H, H-6], 12.53 [s, 1H, NH] ppm; ^13^C NMR (125 MHz, DMSO-*d*_6_) δ 19.4, 35.8, 119.0, 124.6, 126.3, 128.6, 129.1, 129.38, 129.6, 129.6, 129.9, 131.2, 133.4, 134.2, 134.3, 135.3, 135.4, 138.2, 139.5, 144.5, 163.6 ppm. Anal. C_24_H_19_ClF_3_NO_3_S_2_ (525.99); C%: 54.21; H%: 3.40; N%: 2.63, found: C%: 54.21; H%: 3.49; N%: 2.63. HRMS (ESI-TOF) *m/z* calculated for [M + H]^+^ 526.0525, found [M + H]^+^ 526.0440.

##### *N*-[4-Chloro-5-methyl-2-(naphthalen-1-ylmethylthio)benzenesulfonyl]cinnamamide (**16**)

Starting with **6** (0.309 g), the title compound (**16)** was obtained after previous crystallization from ethanol (0.196 g, 55%). Melting point and spectroscopic analysis were in accordance with the previously described in [16]. HRMS (ESI-TOF) *m/z* calculated for [M + H]^+^ 508.0808, found [M + H]^+^ 508.0810.

##### *N*-[4-Chloro-5-methyl-2-(naphthalen-1-ylmethylthio)benzenesulfonyl]cinnamamide (**17**)

Starting with **7** (0.309 g), the title compound (**17**) was obtained after previous purification on silica gel using AcOEt/*n*-hexane (*v*/*v* = 1:3) as an eluent (0.242 g, 68%): m.p. 173.4–175.2 °C; t_R_ = 45.81 min; IR (KBr): 3170 (NH); 3058, 3024 (CH arom); 2922 (CH_3_); 2855 (CH_2_); 1672 (C=O); 1624, 1432 (C=C); 1349, 1159, (SO_2_ sym) cm^−1^; ^1^H NMR (500 MHz, DMSO-*d*_6_): δ 2.33 [s, 3H, CH_3_], 4.59 [s, 2H, CH_2_S], 6.75 [d, 1H, CH=], 7.33 [t, 1H, arom], 7.40 [t, 1H, arom], 7.46–7.55 [m, 5H, arom, C=], 7.58–7.65 [m, 4H, arom], 7.68 [d, 1H, arom], 7.71 [s, 1H, H-3], 7.86 [s, 1H, arom], 7.95 [s, 1H, H-6], 12.53 [s, 1H, NH] ppm. Anal. C_27_H_22_ClNO_3_S_2_ (508.05); C%: 63.59; H%: 4.13; N%: 2.79, found: C%: 63.58; H%: 4.13; N%: 2.78. HRMS (ESI-TOF) *m/z* calculated for [M + H]^+^ 508.0808, found [M + H]^+^ 508.0815.

### 3.2. Bacterial Reference and Clinical Strains

Antimicrobial activity was tested using the following reference strains: *Staphylococcus aureus* ATCC 6538 (MSSA), *Staphylococcus aureus* ATCC 43300 (MRSA), *Staphylococcus epidermidis* PCM 2118, *Escherichia coli* ATCC 11229, *Pseudomonas aeruginosa* ATCC 15442, *Proteus vulgaris* NCTC 4635, *Enterococcus faecalis* ATCC 11420, and *Candida albicans* ATCC 10231.

The antistaphylococcal activity of **10** and **16** was assessed based on 10 clinical non-duplicate *S. aureus* strains isolated from various clinical samples. Five methicillin-resistant *S. aureus* strains were isolated from nose (n = 1), bronchoalveolar lavage (n = 1), sputum (n = 1), wound (n = 1), and blood (n = 1); five methicillin-susceptible *S. aureus* strains were isolated from throat (n = 1), bronchoalveolar lavage (n = 1), sputum (n = 1), abscess (n = 1), and blood (n = 1).

The identity of *S. aureus* isolates was confirmed with conventional methods and with Pastorex Staph-Plus latex agglutination kit (Bio-Rad, Marnes-la-Coquette, France) and verified applying the polymerase chain reaction (PCR) of *S. aureus*-specific region of the thermonuclease gene, *nuc* [20]. Resistance to methicillin was first verified using cefoxitin disk (30 µg), and then, identified based on the presence of *mec*A gene [21]. *S. aureus* ATCC^®^25923 (methicillin-susceptible) and *S. aureus* ATCC^®^43300 (MDR) were applied as the reference strains. Both the reference and clinical strains were stored at −80 °C, in Tryptic Soy Broth (TSB, Becton-Dickinson, Franklin Lakes, NJ, USA) supplemented with 15% glycerol.

### 3.3. Antimicrobial Assay

A broth dilution method was used to determine minimal inhibitory concentration (MIC) for tested compounds as recommended by Clinical Laboratory Standards Institute (CLSI) guidelines [22]. Investigated compounds were diluted in Mueller Hinton Broth 2 (Sigma-Aldrich Co., St. Louis, MO, USA) in polypropylene 96-well plates and initial inoculum 5 × 10^5^ cfu/mL were incubated at 37 °C for 18 h. MIC was taken as the lowest drug concentration at which an observable growth was inhibited. Minimal bactericidal concentration (MBC) was determined in a sample taken from each test tube in which no growth was observed in the MIC assay. The loopful (10 µL) of the tested sample was transferred to Tryptic Soy Agar (TSA, Becton-Dickinson, Franklin Lakes, NJ, USA) and incubated at 37 °C for 48 h. MBC means the lowest concentration of each drug that resulted in more than 99.9% reduction of initial inoculum. Drug Solutions were made fresh on the day of the assay. All experiments were performed in triplicate.

### 3.4. Biofilm Inhibition Assay for **10** and **16**

The assessment of biofilm inhibition for **10** and **16** against selected clinical methicillin-susceptible and methicillin-resistant *S. aureus* strains was conducted by the broth microdilution method in 96-well microtiter plates (Becton-Dickinson, Franklin Lakes, NJ, USA). Briefly, 100 µL of two-fold dilutions of **10** and **16** in TSB at concentrations ranging between 128 µg/mL and 2 µg/mL were delivered into each well. Then, 100 µL bacterial suspension prepared from fresh overnight culture by diluting with TSB to obtain the initial inoculum 2 × 10^6^ CFU/mL was added. The plates were incubated for 24 h at 37 °C. After the incubation, the wells were discarded and were gently rinsed three times with PBS to remove planktonic staphylococcal cells. Then, contents of the wells were discarded and the biofilm was fixed with 200 µL methanol for 20 min and dried at room temperature for 30 min. The biofilm was stained by adding 200 µL 1.8% crystal violet solution for 15 min. After staining, the plates were rinsed with water and 200 µL bleaching solution (30% ethanol, 30% glacial acetic acid, 40% water) added for 15 min to dissolve the remaining crystal violet. A microplater reader was used to measure the absorbance of the adherent biofilm at 570 nm (OD570). For each strain, the result was calculated by subtracting the median OD570 of the triplicates of the negative control from the median OD570 of the triplicates of the samples. The negative control included only the growth medium.

### 3.5. Cell Culture and Cell Viability Assay

All chemicals were obtained from Sigma-Aldrich (St. Louis, MO, USA). The MCF-7 and HeLa cell lines were purchased from Cell Lines Services (Eppelheim, Germany), the HCT-116 cell line was purchased from ATCC (ATCC-No: CCL-247). Cells were cultured in in Dulbecco’s modified Eagle’s medium (DMEM) supplemented with 10% fetal bovine serum, 2 mM glutamine, 100 units/mL penicillin, and 100 μg/mL streptomycin. Cultures were held in an incubator (Heraceus, HeraCell) in a humidified atmosphere with 5% CO_2_ at 37 °C.

Cell viability was evaluated using the MTT (3-(4,5-dimethylthiazol-2-yl)-2,5-diphenyltetrazolium bromide) assay. Stock solutions of the studied compounds were obtained in 100% DMSO. Working solutions were prepared by diluting the stock solutions with DMEM medium, the final concentration of DMSO did not exceed 0.5% in the treated samples. Cells at a density of 5 × 10^3^ cells/well, seeded in 96-well plates, were treated for 72 h with the tested compounds in the concentration range 1–100 μM (1, 10, 25, 50 and 100 μM). After treatment, cells were incubated for 2 h with MTT (0.5 mg/mL) at 37 °C. Cells were lysed with DMSO and the absorbance of the formazan solution was measured at 550 nm with a plate reader (1420 multilabel counter, Victor, Jügesheim, Germany). Values are expressed as the mean ± SD of at least three independent experiments conducted in triplicate.

## 4. Conclusions

Taking into account that the rise of resistant strains has been considered as a current global concern, there is an urgent need to develop new antibiotics. An advantage of new molecules with bacteriostatic or bactericidal activity is a lack of developed resistance in bacteria to these compounds. It is particularly important to find new antibiotics active against *Staphylococcus aureus*, which is a leading cause of death resulting from bacterial infection. In this work we have synthesized a series of *N*-(2-arylmethylthio-4-chloro-5-methylbenzenesulfonyl)amide derivatives including a 2-mercaptobenzenosulfonamide fragment and either cinnamic or cyclohexylpropionic acid residues in their structure. Compounds **8**–**17** showed high bacteriostatic activity against reference staphylococci with low MIC values in the range 4–32 μg/mL. The same low MIC values (4–32 μg/mL) were exhibited by **9**–**12** and **14**–**16** against *E. faecalis*. The most active compounds **10** and **16** also effectively inhibited the growth of ten different strains of *S. aureus* isolated from various human infections (MIC values of 4–8 μg/mL). In the biofilm inhibition tests, we obtained effective antibiofilm agents against four clinical isolates of MRSA and MSSA. Cytotoxic activity assays indicated that compounds with interesting Gram-positive antibacterial activity (**9**–**12**, **14**–**17**) do not show cytotoxicity toward normal cells at concentrations corresponding to the MIC values. The tests revealed also that these compounds do not have cytotoxic activity against cancer cells HCT-116, HeLa, and MCF-7.

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
