# Peer review of "N-(2-Arylmethylthio-4-Chloro-5-Methylbenzenesulfonyl)amide Derivatives as Potential Antimicrobial Agents—Synthesis and Biological Studies"

_ijms, 2019, doi:10.3390/ijms21010210_

Round 1

Reviewer 1 Report

This paper describes the synthesis and anti-microbial screening of a range of sulphonamides which are derivatives of the authors previous work. The chemistry is straightforward and the compounds thoroughly characterised. Screening against reference microorganisms and clinical isolates is comprehensive and the results adequately discussed - two compounds are claimed to show potential for further development. There is scope for future work on more detailed structure-activity investigations. The paper merits publication but requires significant editing. There are a lot of minor grammar and typo errors, which need to be corrected - the required edits are shown below in inverted commas:

Line 17 remove 'the' after show

Line 18 biofilms

Line 19 comma after ..amides

Line 20 'include a' and 'and either cinnamic'

Line 23 biofilms

Line 24 'staphylococci, with the' and 'presenting low MIC'

Line 26 '16 also showed an ability to inhibit biofilms formed'

Line 27 'The potential of'

Line 28 'effect either on'

Line 29 'keratinocytes or human cancer cells, including cervical, colon and breast cancer lines'

Line 37 'years has made'

Line 41 'However, the resistance mentioned above is not'

Line 42 'biofilms'

Line 46 'as well as to provide'

Line 47 'cells, microorganisms'

Line 52 'cells and oxygen'

Line 55 'newer chemotherapies' and 'fight these biofilms'

Line 57 'is a major'

Line 65 'show a wide'

Line 67 'swabs) or intestinal'

Line 68 'among the'

Line 70 '..amides, applying a hybridisation strategy which is a well'

Line 72/73 'with a cinnamic acid residue. The antimicrobial' and 'in several reviews'

Line 75 'permeability and solubility' and 'Within the structure'

Line 76 'incorporated a cyclohexylproprionic acid residue as an aliphatic'

Line 77 'investigate the impact of a saturated versus an aromatic'

Line 84, 85, 88, 89, 93 Italicise the N in the compound name

Line 87 'according to reported'

Line 98 'of a signal'

Scheme 1 and the accompanying text is a bit difficult to follow. I would swap the 'b' and 'c' steps, as I think the sequence of reactions reads more clearly that way. As currently written, b is somewhat superfluous. In the structures, charges of ionic species should be shown.

Lines 107 - 110 All species names should be in italics

Line 111 'allowing the determination of MIC'

Line 115 'expresses the bacteriostatic'

Line 117 ‘to death of’

Line 118/119 ‘quantitatively the potency of an active substance is the MBC value, referring to the minimal concentration of antibiotic at which’

Line 121 Remove comma after growth

Line 123 ‘indicates microorganism tolerance of a given’

Line 129 Candida albicans in italics

Line 132-4 All species in italics

Line 136 ‘worth pointing out’

Line 137 ‘had an ability’

Line 138 E. faecalis in italics. ‘towards some of the tested’

Line 140 ‘low, with MIC values ranging from’

Line 143 ‘positive species have a cell wall’

Line 144 ‘of peptidoglycan layers which may be susceptible to the tested’

Line 145 N in italics

Line 147 ‘presented a low’

Line 148 ‘as well as low’

Line 151 ‘ratio equals 1 against’ S. epidermidis in italics

Line 152 S. epidermidis in italics

Line 153 N in italics and comma after 15

Line 154 ‘by having strong’ S. epidermidis in italics

Line 159 ‘was preserved’

Line 161 ‘values ranging from’

Line 166, 167 S. aureus in italics

Line 170 ‘worth noting’ that’

Line 176 ‘antibiotics, particularly effective in the treatment’

Line 177 ’16 is noteworthy’.

Line 179 ‘These studies’

Lines 181, 189, 191 S. aureus in italics

Line 191 ‘have been’ and ‘etiologic agent of’

Line 193 ‘confers resistance’ and ‘and to the action’

Line 196 ‘at a concentration’

Line 197 ‘of a concentration equal to the MIC’

Line 199 ‘susceptible to compound’

Line 201/202 ‘at a concentration equal to the MIC’

Line 203 comma after mL

Line 204 ‘equals 4 x MIC’ and comma after agent

Line 205 ’16 at a concentration’

Line 206 ‘inhibition of 86%’

Line 207 ‘An ability to inhibit biofilm’

Line 208 ‘as helping to overcome the problem’

Line 210 ‘have been associated with devices’

Line 214 S. aureus in italics

Line 217 ‘if the effect’ and ‘to selective toxicity or a more’

Line 218 ‘we performed an assay on eth human keratinocyte (Ha’

Line 225 ‘do not exhibit’

Line 229 ‘cells, which was higher’ and ‘shows a’

Line 230-232 Species names in italics

Line 236 ‘the absorption range’

Line 244 n-hexane n in italics

Line 264 N in italics in compound name

Line 423 ‘of new molecules with’

Line 424 ‘to these compounds’

Line 425 ‘which is a leading’

Line 427 ‘including a 2-‘

Line 428 ‘and either cinnamic’

Lines 429 and 431 Make bold the compound numbers

Line 431 ‘inhibited the growth’

Line 433 ‘In the biofilm inhibition tests, we’

Line 434 ‘isolates of MRSA’

Complete Funding and Acknowledgements sections

Author Response

Authors thank the reviewer for constructive comments and suggestions that have ultimately improved this manuscript. The manuscript have been modified according to the reviewer's suggestions. Detailed responses to the reviewer are given below.

In the response to Reviewer #1:

We have included all corrections to the manuscript as suggested the Reviewer.

Scheme 1 was improved according to the reviewer suggestion.

Funding section was completed.

Reviewer 2 Report

I only have a few suggestions for the authors prior acceptation of the manuscript, as I think that in general, the results are well described and discussed.

I would suggest that the authors include a discussion of the influence (or the lack of it) in activity of cinnamic acid and cyclohexylpropionic acid in section 2.2. As indicated in the introduction, the authors have included the cyclohexylpropionic acid to study the impact of saturated and aromatic system on bacterial activity, but they have not mentioned this in the results. In table 5, the results for compounds 8 and 13 are not included. I guess this is because they have been already published, but I consider that it would be interesting to include them (with the corresponding reference if required) for comparison. C13 signals for most of the compounds are not included. Please complete funding and acknowledgements sections

Author Response

Authors thank the reviewer for constructive comments and suggestions that have ultimately improved this manuscript. The manuscript have been modified according to the reviewer's suggestions. Detailed responses to the reviewer are given below.

In the response to Reviewer #2:

1. “I would suggest that the authors include a discussion of the influence (or the lack of it) in activity of cinnamic acid and cyclohexylpropionic acid in section 2.2. As indicated in the introduction, the authors have included the cyclohexylpropionic acid to study the impact of saturated and aromatic system on bacterial activity, but they have not mentioned this in the results.”

According to the reviewer suggestion the authors added to the manuscript fragment:

Results of the tests did not indicate significant impact of either aromatic or saturated systems attached to the sulfonamide moiety on antibacterial activity.

2. “In table 5, the results for compounds 8 and 13 are not included. I guess this is because they have been already published, but I consider that it would be interesting to include them (with the corresponding reference if required) for comparison.”

IC50 values for compound 8 were added in Table 5 with the corresponding reference.

Compound 13 were completely inactive in cytotoxic tests on HeLa, HCT-116, MCF-7 cell lines. Growth percent of the treated cells for this compound ranged from 79% to 84% at a concentration of 100 μM and IC50 values were not calculated.  

3. “C13 signals for most of the compounds are not included.”

In the manuscript the authors showed 13C spectra for representative compounds 9 and 15.

4. “Please complete funding and acknowledgements sections”

Funding section was completed.